# Efficient Personalized Adaptation for Physiological Signal Foundation Model

**Chenrui Wu** [1 2]   **Haishuai Wang**[* 1]   **Xiang Zhang** [3]   **Chengqi Zhang** [4]   **Jiajun Bu** [1]

## Abstract

Time series analysis is crucial across various fields like energy, environment, transportation, finance and health. Deep learning has significantly advanced this field, particularly, the Time Series Foundation Model (TSFM) excels in multiple domains due to extensive pre-training. In this work, we focus on TSFM's challenges in medical practice: limited computing resources and medical data privacy. TSFM variants include fine-tuned models and those pre-trained for rapid deployment on diverse data. There may not be enough computing resources to train physiological signals locally in hospitals, and generalized TSFM is still inferior to task-specific methods on private, imbalanced local data. To address this, we propose PhysioPFM, a framework for efficiently personalizing TSFM. Our approach involves low-rank pre-training on public datasets, generator training by trained LoRA weights, and efficient weight generation via local data. Experimental results demonstrate that integrating generated models with TSFM enhances performance, and transferability, and reduces the need for additional sensitive data training.

## 1. Introduction

The recent trend of integrating deep learning algorithms on advanced wearable sensors and fixed medical equipment has catalyzed massive amounts of valuable medical data (Spathis et al., 2020; Che et al., 2017). These recorded medical time series data are continuous observations related to human health, usually including electroencephalogram

[1]Zhejiang Key Laboratory of Accessible Perception and Intelligent Systems, College of Computer Science and Technology, Zhejiang University. [2]School of Computing Science, Simon Fraser University. [3]Department of Computer Science, The University of North Carolina at Charlotte. [4]Department of Data Science and Artificial Intelligence, Hong Kong Polytechnic University. Correspondence to: Haishuai Wang <haishuai.wang@zju.edu.cn>.

*Proceedings of the 42$^{nd}$ International Conference on Machine Learning*, Vancouver, Canada. PMLR 267, 2025. Copyright 2025 by the author(s).

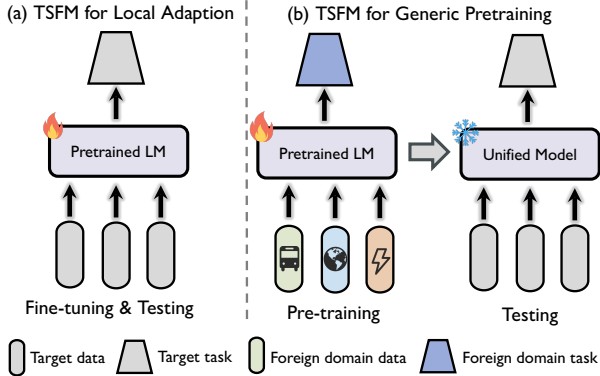

Figure 1. **Illustration of existing two categories of time series foundation model.** (a) represents the methods applying pre-trained LLM to train target data, and conducting tests on target data. (b) demonstrate the methods that pre-train on a large and multi-domain dataset, then test on the target data for the zero-shot prediction.

(EEG), cardiotocography (CTG), electrooculogram (EOG), galvanic skin response (GSR), electrocardiogram (ECG), electromyogram (EMG), and others (Thapa et al., 2024; Wang et al., 2024b; Zhang et al., 2024; Al-Saegh et al., 2021). These physiological signals are usually quantitatively measured by medical devices and then analyzed by doctors or specialists to evaluate the patient's current state and make data-driven decisions, which shows great significance for health monitoring, disease diagnosis, and treatment (Liu et al., 2023). Specifically, we focus on various medical time series classification (TSC) tasks based on the physiological signals, including emotion recognition (Li et al., 2024a), sleep stage detection (Supratak et al., 2017; Dong et al., 2017), neurological disorder classification (Yang et al., 2022b; Zhang et al., 2022), etc.

However, while the community has benefited greatly from the large amount of new data collected by professional medical devices or ubiquitous wearable devices, analyzing physiological signals with existing deep-learning methods still faces inherent challenges in practice. First, the amount of data available for each signal is unbalanced. Most existing studies focus on EEG and ECG data. In contrast, other physiological signals have minor data available, making it challenging to establish unified and generic models for all signals. Furthermore, the sampling frequency and duration of different signals may also vary, further causing the diver-

gence of data features and labels. Therefore, compared with time series in conventional domains, medical time series also have unique challenges in imbalanced distributions.

To effectively capture the complex trends of time series, the foundation model has recently attracted the attention of researchers. With pre-training and success of the language model, diverse time series foundation models (TSFM) have been proposed (Zhou et al., 2023; Jin et al., 2024; Rasul et al., 2023; Liu et al., 2024b; Goswami et al., 2024; Ansari et al., 2024; Garza & Mergenthaler-Canseco, 2023a). TSFM mainly includes two categories as shown in Figure 1. The first category adopts language model parameters as pre-training parameters and performs fine-tuning and testing on target data locally (Zhou et al., 2023; Jin et al., 2024; Rasul et al., 2023; Ansari et al., 2024). Another category pursues generalization performance, is pre-trained with large-scale time series data in various fields, and achieves universal performance of train once for all (Liu et al., 2024b; Goswami et al., 2024; Garza & Mergenthaler-Canseco, 2023a).

Yet, several challenges are aroused by implementing the existing two types of TSFM, shown in Figure 2. For physiological signals, patients' sensitive data is prevented from being uploaded to foundation model service providers for pre-training due to recent laws like EU GDPR for patients' privacy. To adapt local physiological signals, for the first category, in the face of clinical diagnosis scenarios with limited computing capacity, local fine-tuning will cause huge computing overhead, which is quite impractical. If applying the generic pre-trained model, although various foundation models have been proven to perform well in multiple fields and have general feature extraction capabilities, some studies have shown that they perform poorly on specific tasks, such as in fields where publicly available healthcare data is scarce and imbalanced (Glocker et al., 2023; Gupta et al., 2024; Wang et al., 2018a).

While several task-specific TSFMs have been crafted for particular domains, such as Brant-X (Zhang et al., 2024), SleepFM (Thapa et al., 2024) and ECG-FM (McKeen et al., 2024). They aim to fit one task with massive public and private data pre-training. These approaches also face the issue of access to private data for training and the challenge of transferring to unseen physiological signal tasks.

In this work, we propose PhysioPFM, a Personalized Foundation Model approach for Physiological signal to efficiently and lightweight produce a customer model for clinical practice with privacy guarantee. Inspired by the concept Train-Once-for-All Personalization (Chen et al., 2023; Li et al., 2024c), our overall goal is to adapt a TSFM to personalized clinical tasks without extra training during deployment. Based on pre-trained and generalized TSFM, we aim to synthesize customized Low-Rank adapter parameters locally with lightweight cost and privacy guarantee.

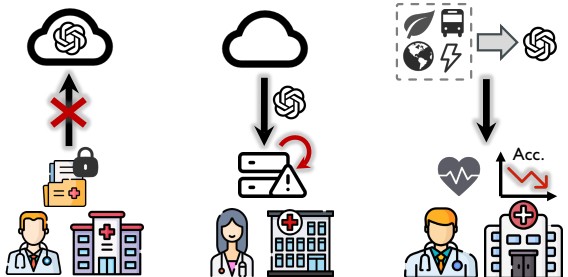

Privacy preservation    Limited comp. resources    Insufficient specialization

*Figure 2.* **The challenges of implementing two types of TSFM to clinical practice.** The first shows that private data cannot be uploaded for pre-training. Local fine-tuning on private data may cause huge computational overhead and be impractical in some conditions. Applying a foundation model pre-trained by multi-domain time series may not fit well with the features and distribution of local physiological signals without adaptation.

Specifically, in the data preparation phase, we first train the TSFM with massive public physiological signal data by Low-Rank Adapter (LoRA) (Hu et al., 2021; Zhou et al., 2024) and store the map of adapters and data sources. To effectively learn diverse features, we adopt the neural collapse (Papyan et al., 2020) to tackle the imbalanced data distribution. In the generator training phase, we use a diffusion transformer (DiT) (Peebles et al., 2022) as a robust generator to synthesize LoRA weights. To bridge the gap between time series and adapter parameters, we capture the representative subsequence of time series shapelets (Wang et al., 2018b; Li et al., 2021; Le et al., 2022; 2024) as the input conditions for DiT training. In the local personalized inference phase, given the local data's shapelets, DiT could generate customized LoRA weights to combine with generic TSFM. A personalized TSFM is obtained for the local medical expert. Only limited computing resources of the generator and TSFM inference are required. We summarize our contributions as follows:

- We novelly propose a personalized approach to transfer the time series foundation model to clinical physiological signal tasks with lower computing costs and privacy guarantees.

- We propose a specialized LoRA generator of DiT, bridging the time series and model by discriminant subsequences shapelets.

- Extensive experiments demonstrate the robustness and efficiency of our PhysioPFM.

## 2. Related Work

### 2.1. Time Series Foundation Model

Recently, the development of large language models (LLMs) has revolutionized the field of natural language process-

ing (NLP) (Chang et al., 2024), demonstrating multimodal adaptability beyond text (Qin et al., 2024; Zeng et al., 2023; Jiang et al., 2024b). Time series have practical value comparable to natural languages, and in essence, they exhibit similar sequentiality in generative modelling (Gruver et al., 2024). This unique feature has prompted much research to adapt LLM to time series (Li et al., 2024b). Current time series foundation models (TSFM) can be divided into two types. The first type utilizes language model parameters to fine-tune target data. llm4ts (Chang et al., 2023) uses fine-tuned transformer modules and positional encodings in GPT-2 to align pre-trained LLMs with time series data for forecasting tasks. One-Fits-All (GPT4TS) (Zhou et al., 2023) freezes the multi-head attention and FFN layers, fine-tunes input, positional embeddings and layer norm blocks, based on GPT-2 parameters. Time-LLM (Jin et al., 2024) introduces a reprogramming architecture, aligning LLM's word embedding with time series embedding, with frozen LLama backbone (Touvron et al., 2023a;b).

Another type targets generalization with large-scale data pretraining for few-shot or zero-shot tasks. LLMTime (Gruver et al., 2024) investigates inputting time series as text with different preprocessing to LLM for zero-shot forecasting. TimeGPT-1 (Garza & Mergenthaler-Canseco, 2023b) is a commercial time series LLM based on the transformer architecture and pre-trained with more than 100 billion data points for zero-shot prediction. Timer (Liu et al., 2024b) proposes a pre-training dataset (UTSD) including multiple domains with 1 billion time points and trains a GPT-style (decoder-only) model to predict. MOMENT (Goswami et al., 2024) also collects a large public dataset: Time Series Pile, then proposes learnable masked patches to train a transformer. These pre-trained TSFMs focus on generalization with a wide domain of time series, where their specialized abilities in medical time series may not be satisfactory. Therefore, we propose a lightweight TSFM personalization scheme to tackle the specialization ability.

### 2.2. Time Series Classification

In general, time series analysis involves four fundamental downstream tasks: forecasting, imputation, classification and anomaly detection. Current TSFMs mostly emphasize forecasting ability or general ability. However, time series classification (TSC) has some unique challenges like heterogeneous timestamp length (Chen et al., 2024), noisy labelling (Liu et al., 2024c), and domain gap from different sensors (He et al., 2023; Wu et al., 2024), aside from only training a robust feature-extracting ability or replacing a prediction head. Therefore, more studies have focused on TSC, especially the applications of physiological signals. For emotion detection, TNAS (Li et al., 2022) combines the Transformer model with NAS and searches through a multi-objective evolutionary algorithm (MOEA) to obtain

the optimal network architecture for EEG-based emotion recognition; OMHGL (Pan et al., 2023) proposes an online multimodal hypergraph learning method for emotion recognition based on multimodal hypergraph fusion and online hypergraph learning. For sleep stage detection, in (Kong et al., 2023), a bilevel optimization approximation for EEG-based sleep stage classification is proposed with a neural architecture search framework; NeuroNet (Lee et al., 2024) proposes a self-supervised learning framework for sleep state detection by integrating contrastive learning tasks and masked prediction tasks. To uniformly solve the physiological signal classification, Bran-X (Zhang et al., 2024) utilizes a pre-trained brain signal foundation model, Brant-2 (Yuan et al., 2024) with 2 TB private data, then aligns other signals to brain signals in two-level patching. In this work, we aim to efficiently transfer the general knowledge of TSFM to physiological signals with public datasets. Our focus is to achieve specialization through openness and protect privacy without exposing patients' private data. We also consider the classification challenges of noisy and imbalanced time series in physiological signal data.

## 3. Preliminaries

### 3.1. Problem Setup

We consider multi-class time series classification tasks under a train-once-for-all personalization scenario (Chen et al., 2023; Li et al., 2024c). There is a primitive time series foundation model $W_0$ pre-trained by large-scale public datasets $\mathcal{D}$. Each local clinical expert/user $k$ holds a private dataset $\mathcal{D}_k = \{(x_i^k, y_i^k)\}_{i=1}^{n_k}$, where $x_i^k$ is the input of the training sample, corresponding $y_i^k$ denotes the given label, and the sum of local data does not belong to pre-training datasets $\bigcup_{i=1}^{k} \mathcal{D}_i \cap \mathcal{D} = \varnothing$. For each physiological signal $x \in \mathbb{R}^{M \times L}$ includes $M$ channels with a length of $L$ timestamps. The optimization objective is to obtain a best personalized model $W_k'$ for each user based on private dataset $\mathcal{D}_k$, formulated as:

$$\min \mathcal{L}(W_k') = \mathcal{L}_{(}f_\phi(W_0, \mathcal{D}_k)), \qquad (1)$$

where $f_\phi$ denote the personalization operation function based on the generative network $G_\phi$.

### 3.2. Low-Rank Adaptation

The size of the parameters of existing LLM usually starts from billions. Many works (Li et al., 2018) have proved that matrices for deep learning are often over-parameterized. Low-Rank Adaptation (LoRA) (Hu et al., 2021) allows indirect training of some dense layers in a neural network by optimizing the rank factorization matrix of the dense layers that change during the adaptation process while keeping the pre-trained weights unchanged. Hence, LoRA improves efficiency for fine-tuning large pre-trained language

models. In detail, LoRA first fixes the pre-trained LLM $W_0 \in \mathbb{R}^{d \times k}$ and introduces two trainable low-rank matrices: dimensionality-reducing matrices $A \in \mathbb{R}^{r \times k}$ and dimension-raising matrix $B \in \mathbb{R}^{d \times r}$, where $r \ll \min(d, k)$. The LoRA's forward pass equation is denoted as:

$$W_0 x + \Delta W x = W_0 + BAx, \quad (2)$$

where $\Delta W = BA$ is the parameter needing to be updated, $x$ is the same input for $W_0$ and $\Delta W$. The more concentrated and stable parameter distribution in LoRA helps improve generalization by reducing the risk of overfitting. During the inference process, LoRA also introduces almost no additional inference latency. Only the calculation of $W = W_0 + \Delta W$ is required. In this work, we aim to synthesize LoRA parameters to directly obtain customized LLMs based on target data in a local private diagnosis.

### 3.3. Neural Collapse

Neural collapse is a phenomenon in which the feature prototypes and weights vectors of the classifier will gradually converge to a Simplex Equiangular Tight Frame (ETF) structure at the terminal phase of balanced data training (Papyan et al., 2020; Li et al., 2023; Yang et al., 2023; 2022a). We note the definition of the ETF and Neural Collapse as follows:

**Definition 3.1** (Simplex Equiangular Tight Frame). A set of $C$-dimensional vectors $\mathbf{w}_i \in \mathbb{R}^d$, where $i \in [C]$ and $d \geq C - 1$, forms a simplex equiangular tight frame (ETF) if it satisfies the condition:

$$\mathbf{W} = \sqrt{\frac{C}{C-1}} \mathbf{U} \left( \mathbf{I}_C - \frac{1}{C} \mathbf{1}_C \mathbf{1}_C^T \right), \quad (3)$$

where $\mathbf{W} = [\mathbf{w}_1, \cdots, \mathbf{w}_C] \in \mathbb{R}^{d \times C}$, $\mathbf{U} \in \mathbb{R}^{d \times C}, (d \geq C)$ is a partial orthogonal matrix satisfying $\mathbf{U}^T \mathbf{U} = \mathbf{I}_C$. $\mathbf{I}_C \in \mathbb{R}^{C \times C}$ is a identity matrix and $\mathbf{1}_C$ is a vector of ones. Any column vectors in $\mathbf{W}$ has the same $\ell_2$ norm, and the pair-wise angles between any pair of vectors are maximized by the simplex ETF, denoted as:

$$\mathbf{w}_i^T \mathbf{w}_j = \frac{C}{C-1} \delta_{i,j} - \frac{1}{C-1}, \forall i, j \in [C], \quad (4)$$

where it takes the value of 1 if the indices $i$ and $j$ are equal, while yielding 0 in all other cases. The optimal condition could be obtained when $\forall i \neq j$, the maximal equiangular separation is $\mathbf{w}_i^T \mathbf{w}_j = -\frac{1}{C-1}$.

The phenomenon of Neural Collapse encompasses four distinct properties, which we highlight as key manifestations:

- **NC1 (Within-Class Variability Collapse):** All the last-layer features will collapse to the class prototypes (within-class feature means). Formally, the covariance $\Sigma_\theta^c \to \mathbf{0}$, and $\Sigma_\theta^c := \frac{1}{n_c} \sum_{i=1}^{n_c} (\mathbf{h}_{c,i} - \mathbf{h}_c)(\mathbf{h}_{c,i} - \mathbf{h}_c)^T$, where $n_c$ is the index of samples in $c$-th class, $\mathbf{h}_{c,i} = \mathbb{F}(\theta; x_i)$ is the feature of the $i$-th sample of $c$-th class, and $\mathbf{h}_c = \frac{1}{n_c} \sum_{i=1}^{n_c} \mathbf{h}_{c,i}$ is the within-class mean of class $c$ features, that is the prototype.

- **NC2 (Convergence to Simplex ETF):** The class prototypes will collapse to a simplex ETF and be maximally separated. $\hat{\mathbf{h}}_c = (\mathbf{h}_c - \mathbf{h}_G)/||\mathbf{h}_c - \mathbf{h}_G||, \forall c \in [C]$ satisfies Eq. (4), where $\mathbf{h}_G$ refers to the global mean of all the features.

- **NC3 (Convergence to Self-Duality):** The classifier weights will collapse to the same simplex ETF of the corresponding class. $\hat{\mathbf{h}}_c = \mathbf{w}_c/||\mathbf{w}_c||$, where $\mathbf{w}_c$ is the classifier weight vector of class $c$.

- **NC4 (Simple Decision Rule):** Under the conditions of **(NC1)-(NC3)**, the classifier's prediction can be simplified to the nearest class centers based on Euclidean distances, denoted as: $\mathrm{argmax}_k \langle \mathbf{h}, \mathbf{w}_c \rangle = \arg\min_k ||\mathbf{h} - \mathbf{w}_c||$, where $\mathbf{h}$ represents the last-layer feature of a sample used for classification.

## 4. Proposed Method

### 4.1. Overview

As shown in Figure 3, we describe the detailed process of the proposed physiological signal foundation model personalization method: PhysioPFM. We consider three main processes: debiased alignment as data preparation, shapelet-to-LoRA model training, and lightweight custom model generation. First, we adopt public physiological signals to fine-tune the original time series foundation model in a LoRA way. Then we use the map of the low-rank adapter and the corresponding data to train the generative model. Finally, the user could apply the generative model to produce a customer low-rank adapter with private data. Combined with the time series foundation model, a customized physiological signal foundation model is obtained with only the extra computing costs of generation.

### 4.2. Data Preparation

Specifically, with the wide success of Low-Rank Adapter (LoRA) fine-tuning in foundation models (Hu et al., 2021; Zhou et al., 2024), we first pre-train the time series foundation model with plenty of public physiological signal datasets in a LoRA way and store the map of adapters and data sources. To synthesize learnable LoRA parameters, we consider building a map between LoRA and input time series by training the generative model. We adopt multiple public physiological signal datasets $\mathcal{D}^{pub}$ and decompose

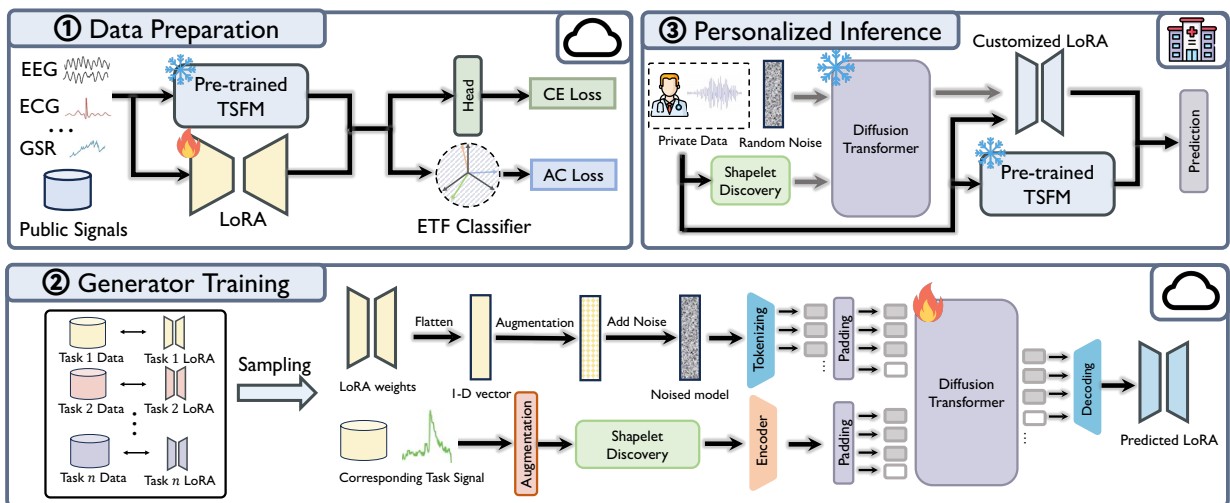

*Figure 3.* **The framework of the proposed PhysioPFM.** The training procedure follows: ① In the data preparation phase, we pre-train a time series foundation model by public physiology signals in the cloud, to obtain the map of data and LoRA weights.② In the generator training phase, we train a robust weights generator by the data as a condition and LoRA weights as output. ③ In the personalized inference phase, we adopt local data and the trained generator to synthesize customer LoRA weights for the TSFM.

them into more sub-tasks $\mathcal{D}_k^{pub}$ to expand the amount of training tasks. For each task, we can obtain a personalized LoRA. Given the forward pass of LoRA in Eq. 2, we adopt two classification losses to update the original TSFM and get a customized LoRA for every task $t_k$.

**Cross-Entropy Loss.** The vanilla cross-entropy loss $\mathcal{L}_{ce}$ is used to minimise empirical risks in a classification task:

$$\mathcal{L}_{ce} = -\frac{1}{|\mathcal{D}_k^{pub}|} \sum_{i=1}^{|\mathcal{D}_k^{pub}|} y_i \log(f(W_0; x_i)), \quad (5)$$

where $\mathcal{D}_k^{pub}$ is the subset of the public physiological signal dataset $\mathcal{D}^{pub}$. To enrich the training data size, each individual dataset and its subset of spatial classes can be $\mathcal{D}_k^{pub}$. The specific public physiological signal datasets we collected are provided in Appendix A.6.

**Anchored Classifier Loss.** Due to the class-level distribution imbalance and length heterogeneity of the collected physiological signal dataset, the number and series length may differ across classes and samples. We aim to update the TSFM in a debiased way. We first define the prototype $p_i$ as the mean of embedding of the class $i \in \{1, 2, \cdots, C\}$ (Snell et al., 2017). According to Definition 1, we can derive an optimal condition of class prototypes:

**Proposition 4.1.** *In a multi-class classification problem, if prototypes $p_1 \cdots p_C \in \mathbb{R}^d$ are fixed and satisfy $p_i^T p_j = -\frac{1}{C-1}, \forall i \neq j, i, j \in [C]$, the maximum of angular between any two class prototypes will be reached, where the decision bound among classes is optimal.*

This proposition provides the optimality condition of vectors to maximize the minimal sample margin. Thus, we initialize

a random synthesis of a simplex equiangular tight frame $\hat{\mathbf{p}} \in \mathbb{R}^{d \times C}$. Any pair of prototypes vectors $\mathbf{p}_i, \mathbf{p}_j$ in $\hat{\mathbf{P}}$ satisfy a maximal pair-wise separation $\mathbf{p}_i^T \mathbf{p}_j = -\frac{1}{C-1}, \forall i \neq j, i, j \in [C]$. Therefore, we add the classifier-anchored loss to regularize the training, defined as:

$$\mathcal{L}_{ac} = -\log \frac{n_{y_i} \exp(x_i \cdot \mathbf{p}_{y_i}/\tau)}{\sum_{y_j \in \{[C] \setminus y_i\}} n_{y_j} \exp(x_i \cdot \mathbf{p}_{y_j}/\tau)}, \quad (6)$$

where $n_{y_i}$ is the number of samples for class $y_i$, $\tau$ is a temperate hyper-parameter, we set $\tau = 1$ generally. Through these losses, the TSFM could be adapted to various physiological signal sub-tasks with a debiased classifier. The overall loss function to align the generic TSFM to physiological signals by low-rank adaptation is formulated as:

$$\mathcal{L} = \mathcal{L}_{ce} + \mathcal{L}_{ac}. \quad (7)$$

After LoRA training on each sub-task of public datasets, we define the sub-task of $\mathcal{D}_k^{pub}$ as $t_k$. The overall demonstration of the obtained data sample map can be expressed as $\mathcal{T} = \{(\Delta W_k, t_k)\}_{k=1}^K, T_k \in \mathcal{T}$, where $\Delta W_k = B_k A_k$.

### 4.3. Generator Training

To train the parameter generator, model generation with advances in diffusion process (Peebles et al., 2022; Jiang et al., 2024a) recently stands out from traditional techniques like network architecture search (Annavajjala et al., 2024; Cai et al., 2018) and neural network compression (Han et al., 2015; Wang et al., 2022), beyond image, video and other multimodal tasks. Different from generating a whole backbone model like a CNN and an MLP (Peebles et al., 2022; Li et al., 2024c), LoRA's size of parameters shows a

minor feature space and is easier to compress and predict by controlling the rank of the adapter $r$. As a more efficient paradigm for generating weights, we present the details of DiT training.

**Input learnable parameters.** Through the data preparation in Section 4.2, we could obtain the pairs $T_k = (\Delta W_k, t_k)$. We set LoRA weights $\Delta W_k$ as input, flatten all parameters into a one-dimensional vector and chunk/tokenize the parameters in each layer. We add augmentation and random noise while inputting.

**Input conditions.** Existing works (Chen et al., 2023; Li et al., 2024c) build the connection between model weights and textual prompts for image classification tasks by powerful pre-trained CLIP (Radford et al., 2021). However, physiological signals are chronological data points and no such existing encoder bridges the text-to-time relationships as CLIP. To compress the input space and bridge the time series with LoRA weights, we propose shapelet prototypes as a substitute. Shapelet is a set of discriminative subsequences of a time series (Ye & Keogh, 2009), each of which is expected to represent a class best. Therefore, shapelet can help to better understand the meaning of the time series and is widely used in the field of time series classification (Li et al., 2021; Zhang & Sun, 2022; Liu et al., 2024d). Following the existing works (Le et al., 2022; 2024), we slice the time series by the Offline shapelet Discovery (OSD) method to get the shapelets $\mathbf{S}_i(t_i)$, presented in Appendix A.1. For each class, we set a uniform amount of shapelets.

**Diffusion training.** We adopt DiT as the generator to train. We present the basic training process of the DiT model as:

**Diffusion forward pass.** The forward process is adding noise to the initial sample from $x_{t-1}$ to $x_t$, by condition $\mathbf{c}$. The probability of transitioning is modeled as a Gaussian distribution:

$$q(x_t|x_{t-1}, \mathbf{c}) = \mathcal{N}(x_t; \sqrt{1 - \beta_t}x_{t-1}, \beta_t \mathcal{I}) \quad (8)$$

where $\beta_t$ are the timestep-dependent noise levels, and $\mathbf{I}$ represents the identity matrix. The complete forward process is denoted as:

$$q(x_{1:T}|x_0, \mathbf{c}) = \prod_{t=1}^{T} q(x_t|x_{t-1}, \mathbf{c}) \quad (9)$$

**Diffusion reverse process.** The reverse process reconstructs the original sample from the noisiest state $x_T$ with condition $\mathbf{c}$:

$$p_\phi(x_{t-1}|x_t, \mathbf{c}) = \mathcal{N}(x_{t-1}; \mu_\phi(x_t, t, \mathbf{c}), \Sigma_\phi(x_t, t, \mathbf{c})) \quad (10)$$

**Diffusion training objective.** The training objective to predict corresponding LoRA weights is to minimize the simplified variational lower bound, denoted as:

$$\min_\phi \mathcal{L}(\phi) \sum_{k \in K} \sum_{j \in \mathcal{J}} ||\Delta W_k - G_\phi(\mathbf{S}(t_k), \Delta W_k^j)||_2^2, \quad (11)$$

where $G_\phi$ is the generator model, $\mathbf{S}(t_k)$ is the shapelets of task $t_k$ inputted as conditions, noised LoRA weights is $\Delta W_k^j$. $j \in [J]$ denotes the timestep in the diffusion forward noising process. The learning objective of diffusion is to minimize the simplified variational lower bound, which reduces to predicting the denoised model parameters.

### 4.4. Personalized Inference

Given the trained generator network $G_\phi$, we could deploy on local medical users. The user provides raw data to the encoder to obtain shapelets as conditions. With the input conditions, the generator $G_\phi$ could synthesize suitable low-rank adapter weights $\Delta W'$. Combined with the original TSFM, the expert could conduct a diagnosis of disease by updated TSFM, which only requires the cost of the generator and foundation model inference. The advantage of our personalized inference lies in these aspects: 1) Lower computing overhead, compared with other works, in local deployment, we only add the cost of the generating process instead of costly computation of foundation model fine-tuning. 2) Privacy preservation, our method does not need to upload sensitive data to the server for pre-training, and the training for the DiT is also based on public physiological signals. DiT's powerful generalization capability enables the entire personalized model generation process to be performed locally. 3) The best of two worlds with public and private data. Our data preparation and generator training depend on public data, but our method can adapt to private data. The full utilization of both data boosts the overall performance.

## 5. Experiments

### 5.1. Experimental Setup

**Datasets and tasks** To fully evaluate performance, we consider four typical physiological signals classification tasks, covering different signals. We randomly sample 60% of the data for training, 20% for validation, and 20% for testing, following the existing work (Zhang et al., 2024), for all evaluation tasks. The specific datasets are as follows, and the introduction of each task is shown in Appendix A.2.

*Sleep-state detection.* Sleep-EDF dataset (Kemp et al., 2000) is a public dataset containing multiple sleep records. These records are collected through multi-channel physiological signals including EEG, EOG, and EMG. For Sleep-EDF, we consider five class sleep stages: Wake, N1 (Light sleep stage 1), N2 (Light sleep stage 2), N3 (Deep sleep stage), and REM (Rapid Eye Movement) sleep. Parts of previous work

*Table 1.* **Results in terms of test accuracy (%) performance on Sleep-state detection.** Blue/**bold** fonts highlight the best baseline/our method. We evaluate classification accuracy, macro F1 score and kappa value.

| Method | Overall Metrics | | | Perclass F1 Score | | | | |
|---|---|---|---|---|---|---|---|---|
| Metrics | Acc. | MF1 | $\kappa$ | Wake | N1 | N2 | N3 | REM |
| Informer[2022] | $71.91_{\pm1.82}$ | $69.93_{\pm1.17}$ | $55.28_{\pm2.25}$ | $65.63_{\pm1.77}$ | $52.85_{\pm3.01}$ | $52.37_{\pm2.08}$ | $68.76_{\pm1.89}$ | $53.82_{\pm2.30}$ |
| FEDformer[2022] | $56.58_{\pm2.39}$ | $50.35_{\pm2.12}$ | $51.78_{\pm1.02}$ | $60.50_{\pm3.25}$ | $40.89_{\pm1.74}$ | $55.53_{\pm3.34}$ | $49.13_{\pm2.63}$ | $46.52_{\pm2.15}$ |
| SimMTM[2023] | $65.39_{\pm1.40}$ | $55.53_{\pm1.92}$ | $52.47_{\pm2.84}$ | $63.90_{\pm2.22}$ | $38.44_{\pm1.67}$ | $50.73_{\pm2.19}$ | $58.48_{\pm3.53}$ | $66.89_{\pm2.30}$ |
| TimesNet[2023] | $77.95_{\pm2.67}$ | $70.28_{\pm2.42}$ | $68.36_{\pm1.83}$ | $80.48_{\pm2.91}$ | $57.24_{\pm2.78}$ | $70.45_{\pm1.65}$ | $68.39_{\pm3.10}$ | $75.73_{\pm3.28}$ |
| PatchTST[2023] | $74.51_{\pm1.30}$ | $68.19_{\pm2.53}$ | $66.87_{\pm2.47}$ | $77.24_{\pm2.39}$ | $56.04_{\pm2.85}$ | $66.30_{\pm4.10}$ | $70.35_{\pm2.34}$ | $71.24_{\pm2.97}$ |
| iTranformer[2024] | $76.43_{\pm2.47}$ | $63.49_{\pm1.77}$ | $68.20_{\pm2.91}$ | $79.38_{\pm2.68}$ | $53.05_{\pm2.43}$ | $65.32_{\pm2.73}$ | $60.45_{\pm1.59}$ | $58.06_{\pm3.61}$ |
| OneFitsAll[2023] | $71.86_{\pm3.16}$ | $65.04_{\pm1.51}$ | $63.28_{\pm2.33}$ | $80.24_{\pm3.09}$ | $49.17_{\pm2.54}$ | $68.49_{\pm2.02}$ | $62.47_{\pm2.35}$ | $66.98_{\pm3.49}$ |
| Time-LLM[2023] | $80.95_{\pm2.68}$ | $71.22_{\pm2.34}$ | $72.63_{\pm1.52}$ | $82.56_{\pm2.45}$ | $59.38_{\pm2.23}$ | $65.82_{\pm3.52}$ | $73.40_{\pm2.31}$ | $76.38_{\pm2.90}$ |
| MOMENT[2024] | $81.90_{\pm1.42}$ | $72.72_{\pm2.82}$ | $73.12_{\pm1.17}$ | $78.62_{\pm2.88}$ | $54.69_{\pm2.56}$ | $77.07_{\pm1.89}$ | $76.34_{\pm2.72}$ | $75.05_{\pm4.11}$ |
| SleepDG[2024] | $79.48_{\pm3.16}$ | $71.95_{\pm2.30}$ | $73.43_{\pm2.44}$ | $81.29_{\pm2.72}$ | $64.52_{\pm1.03}$ | $72.30_{\pm2.45}$ | $69.24_{\pm2.93}$ | $69.75_{\pm3.42}$ |
| SleepFM[2024] | $81.65_{\pm2.05}$ | $74.26_{\pm3.40}$ | $72.03_{\pm3.97}$ | $82.55_{\pm2.93}$ | $61.24_{\pm2.56}$ | $76.52_{\pm3.15}$ | $74.38_{\pm2.27}$ | $76.93_{\pm2.04}$ |
| **Our PhysioPFM** | $\mathbf{86.39_{\pm2.54}}$ | $\mathbf{76.27_{\pm2.98}}$ | $\mathbf{77.36_{\pm3.14}}$ | $\mathbf{83.04_{\pm2.75}}$ | $\mathbf{66.71_{\pm2.81}}$ | $76.43_{\pm1.94}$ | $\mathbf{77.29_{\pm2.90}}$ | $\mathbf{78.38_{\pm2.31}}$ |

adopt 30-minute data before and after in-bed, for fairness, we do not adopt this additional cropping, just raw data.

*Emotion detection.* DREAMER (Katsigiannis & Ramzan, 2017) is a multimodal physiological signal library consisting of EEG and ECG signals recorded during emotional arousal via audiovisual stimulation. Signals were recorded from 23 participants, along with the participants' self-assessment of their affective state after each stimulation, including valence, arousal, and dominance.

*Arrhythmia diagnosis.* The MIT-BIH arrhythmia dataset (Moody & Mark, 2001) includes more than 4,000 24-hour periodic dynamic ECG signals of 47 test individual units, 48 recording files with a duration of approximately 30 minutes, and a total of 109,500 heartbeats, of which abnormal heartbeats account for approximately 30%.

*Freezing of Gaits Detection.* The FOG dataset (Li, 2021) collects multimodal data including EEG, EMG, ECG, skin conductivity (SC), and acceleration (ACC) during walking tasks using a high-quality hardware system that integrates commercially available and self-designed sensors. A total of 12 PD patients completed standard hospital experiments.

**Baselines.** We take three groups of methods as baselines. (1) vanilla deep learning technique for time series: SimMTM (Dong et al., 2023), TimesNet (Wu et al., 2023), iTtransformer (Liu et al., 2024a), PatchTST (Nie et al., 2023), FEDformer (Zhou et al., 2022), Informer (Zhou et al., 2021). (2) time series foundation model: OneFitsAll (Zhou et al., 2023), Time-LLM (Jin et al., 2024), MOMENT (Goswami et al., 2024). (3) for each downstream task, we adopt specialized methods: SleepFM (Thapa et al., 2024) and SleepDG (Wang et al., 2024a) for sleep state detection, LSTM-MLP (Wang et al., 2023) and OMHGL (Pan et al., 2023) for emotion detection, DeepArr (Midani et al., 2023) for arrhythmia detection, Extra Tree Classifier (Goel et al., 2023) for FoG task.

**Implement details.** For generative model architecture, we adopt GPT-2 (Radford et al., 2019) as the diffusion transformer with 12 layers. During training, we use AdamW with a batch size of 64, a learning rate of 4e-4, 1000 diffusion steps, and a linear noise scheduler ranging from 0.0001 to 0.012. we divide the LoRA weights into chunks by layer, and the size of each chunk is 576. We set the rank of the adapter as 4. For the pre-trained time series foundation model, we adopt the 6-layer GPT2-based backbone (Radford et al., 2019; Liu et al., 2024b), pre-trained by UTSD datasets (Liu et al., 2024b). We collect public physiological signal datasets for training in the data preparation process, shown in Table 6. The target task's data will be removed from generator training. We further prepare subsets and sub-classes of datasets to expand the number of tasks. We adopt the average result of 3 times.

## 5.2. Main Results

In Table 1-4, we evaluate the performance of our proposed method against diverse baseline methods including the general deep learning time series model, time series foundation model with local training, pre-trained time series model for zero-shot deployment and specialized methods for downstream tasks. The results show that PhysioPFM consistently outperforms all baseline methods on all four tasks, demonstrating the remarkable personalization ability of the proposed method. Specifically, for sleep state detection in Table 1, PhysioPFM achieves 4.49% surpassing the best baselines MOMENT. PhysioPFM also leads with the best performance in the F1 score and is stable in the classification ability for each class, even in the tough class N1 light sleep stage, with a 2.2% advantage.

For the evaluation of DREAMER datasets in Table 2, our PhysioPFM method achieves the best results in emotion recognition with the averaged accuracy of 73.97%, 82.19%

*Table 2.* **Results in terms of test accuracy (%) performance on emotion detection.** Blue/**bold** fonts highlight the best baseline/our method. We evaluate the accuracy, F1 score and AUC.

| Dataset | Valence | | | Arousal | | | Dominance | | |
|---|---|---|---|---|---|---|---|---|---|
| Metrics | Acc. | F1 | AUC | Acc. | F1 | AUC | Acc. | F1 | AUC |
| Informer$_{2022}$ | 62.93$_{\pm2.85}$ | 70.38$_{\pm4.08}$ | 65.53$_{\pm2.18}$ | 75.20$_{\pm3.12}$ | 80.87$_{\pm2.78}$ | 77.39$_{\pm4.77}$ | 76.30$_{\pm2.30}$ | 85.42$_{\pm2.83}$ | 83.25$_{\pm3.29}$ |
| FEDformer$_{2022}$ | 55.30$_{\pm3.78}$ | 63.29$_{\pm2.85}$ | 60.53$_{\pm2.93}$ | 70.34$_{\pm4.34}$ | 75.85$_{\pm2.91}$ | 72.56$_{\pm3.66}$ | 72.26$_{\pm2.78}$ | 81.19$_{\pm3.34}$ | 80.04$_{\pm3.96}$ |
| SimMTM$_{2023}$ | 60.89$_{\pm2.54}$ | 75.53$_{\pm3.40}$ | 67.20$_{\pm3.92}$ | 76.64$_{\pm2.89}$ | 83.49$_{\pm3.73}$ | 77.24$_{\pm3.51}$ | 76.44$_{\pm2.07}$ | 86.56$_{\pm3.87}$ | 82.17$_{\pm2.88}$ |
| TimesNet$_{2023}$ | 70.32$_{\pm2.87}$ | 78.98$_{\pm3.38}$ | 75.35$_{\pm4.23}$ | 78.95$_{\pm3.15}$ | 85.67$_{\pm3.59}$ | 78.19$_{\pm2.84}$ | 81.26$_{\pm2.54}$ | 85.07$_{\pm3.27}$ | 84.72$_{\pm3.98}$ |
| PatchTST$_{2023}$ | 62.35$_{\pm3.14}$ | 72.80$_{\pm4.05}$ | 66.73$_{\pm2.27}$ | 71.29$_{\pm3.63}$ | 82.24$_{\pm2.70}$ | 75.54$_{\pm3.32}$ | 75.47$_{\pm2.85}$ | 82.54$_{\pm3.61}$ | 79.93$_{\pm4.41}$ |
| iTranformer$_{2024}$ | 67.93$_{\pm4.27}$ | 77.42$_{\pm2.38}$ | 75.60$_{\pm3.88}$ | 80.59$_{\pm2.72}$ | 84.05$_{\pm5.45}$ | 80.82$_{\pm3.76}$ | 80.20$_{\pm4.17}$ | 83.91$_{\pm2.10}$ | 82.94$_{\pm3.98}$ |
| OneFitsAll$_{2023}$ | 64.04$_{\pm2.32}$ | 73.43$_{\pm3.03}$ | 64.23$_{\pm2.22}$ | 75.03$_{\pm4.28}$ | 81.29$_{\pm5.14}$ | 72.39$_{\pm2.05}$ | 77.46$_{\pm3.88}$ | 85.09$_{\pm4.29}$ | 83.40$_{\pm3.46}$ |
| Time-LLM$_{2023}$ | 67.49$_{\pm2.93}$ | 70.35$_{\pm3.89}$ | 76.29$_{\pm3.02}$ | 77.32$_{\pm4.66}$ | 84.05$_{\pm3.85}$ | 74.21$_{\pm3.39}$ | 80.49$_{\pm2.03}$ | 83.35$_{\pm3.49}$ | 84.31$_{\pm4.76}$ |
| MOMENT$_{2024}$ | 69.20$_{\pm3.85}$ | 77.34$_{\pm4.40}$ | 74.22$_{\pm2.68}$ | 80.51$_{\pm2.87}$ | 83.46$_{\pm2.56}$ | 75.32$_{\pm4.39}$ | 79.39$_{\pm5.30}$ | 86.56$_{\pm3.02}$ | 86.90$_{\pm2.45}$ |
| LSTM-MLP$_{2023}$ | 67.60$_{\pm3.62}$ | 78.36$_{\pm4.61}$ | 62.21$_{\pm2.06}$ | 77.78$_{\pm3.37}$ | 84.90$_{\pm3.92}$ | 78.24$_{\pm4.74}$ | 79.11$_{\pm2.69}$ | 89.63$_{\pm5.53}$ | 82.55$_{\pm3.10}$ |
| OMHGL$_{2023}$ | 71.29$_{\pm3.13}$ | 66.20$_{\pm2.95}$ | 70.09$_{\pm2.29}$ | 79.53$_{\pm2.62}$ | 79.21$_{\pm3.41}$ | 78.83$_{\pm4.74}$ | 79.41$_{\pm2.35}$ | 83.52$_{\pm4.30}$ | 80.69$_{\pm1.36}$ |
| **Our PhysioPFM** | **73.97$_{\pm2.36}$** | **79.25$_{\pm2.56}$** | **79.41$_{\pm1.94}$** | **82.19$_{\pm2.73}$** | **86.65$_{\pm4.05}$** | **80.84$_{\pm3.16}$** | **84.45$_{\pm4.35}$** | **91.53$_{\pm2.82}$** | **89.61$_{\pm3.15}$** |

*Table 3.* **Results in terms of model performance on arrhythmia diagnosis.** We evaluate accuracy, precision, recall and F1.

| Methods/Metrics | Acc. | Prec. | Rec. | F1 |
|---|---|---|---|---|
| Informer | 77.54$_{\pm2.30}$ | 74.20$_{\pm2.79}$ | 78.34$_{\pm3.42}$ | 76.21$_{\pm2.39}$ |
| FEDformer | 60.35$_{\pm1.95}$ | 55.46$_{\pm3.44}$ | 60.92$_{\pm2.18}$ | 58.06$_{\pm2.57}$ |
| SimMTM | 80.27$_{\pm3.49}$ | 82.37$_{\pm2.35}$ | 79.50$_{\pm3.52}$ | 80.90$_{\pm2.19}$ |
| TimesNet | 81.33$_{\pm2.18}$ | 78.93$_{\pm2.67}$ | 82.45$_{\pm1.72}$ | 80.65$_{\pm2.06}$ |
| PatchTST | 70.53$_{\pm3.41}$ | 73.65$_{\pm2.78}$ | 67.45$_{\pm4.26}$ | 70.41$_{\pm2.85}$ |
| iTranformer | 75.56$_{\pm2.96}$ | 78.95$_{\pm2.67}$ | 74.41$_{\pm3.68}$ | 76.61$_{\pm2.94}$ |
| OneFitsAll | 75.63$_{\pm3.04}$ | 71.90$_{\pm2.34}$ | 74.45$_{\pm3.81}$ | 73.15$_{\pm2.79}$ |
| Time-LLM | 82.91$_{\pm1.71}$ | 84.22$_{\pm3.45}$ | 81.02$_{\pm2.51}$ | 82.58$_{\pm2.90}$ |
| MOMENT | 83.25$_{\pm2.09}$ | 86.92$_{\pm2.31}$ | 83.57$_{\pm2.82}$ | 85.21$_{\pm2.69}$ |
| DeepArr | 86.55$_{\pm2.62}$ | 83.34$_{\pm2.75}$ | 85.95$_{\pm3.18}$ | 84.62$_{\pm2.71}$ |
| **Ours PhysioPFM** | **89.94$_{\pm1.53}$** | **88.42$_{\pm3.19}$** | **90.72$_{\pm2.81}$** | **89.55$_{\pm2.40}$** |

*Table 4.* **Results of model performance on Freezing of Gaits Detection.** We evaluate accuracy, precision, recall and F1.

| Methods/Metrics | Acc. | Prec. | Rec. | F1 |
|---|---|---|---|---|
| Informer | 64.25$_{\pm1.85}$ | 66.34$_{\pm2.56}$ | 74.21$_{\pm2.96}$ | 70.05$_{\pm2.32}$ |
| FEDformer | 59.80$_{\pm3.44}$ | 57.67$_{\pm2.85}$ | 68.15$_{\pm4.09}$ | 62.47$_{\pm3.48}$ |
| SimMTM | 70.61$_{\pm2.20}$ | 69.25$_{\pm3.71}$ | 72.47$_{\pm5.23}$ | 70.82$_{\pm3.72}$ |
| TimesNet | 72.19$_{\pm2.26}$ | 70.88$_{\pm3.56}$ | 73.79$_{\pm2.74}$ | 72.30$_{\pm3.47}$ |
| PatchTST | 66.90$_{\pm4.42}$ | 67.23$_{\pm5.71}$ | 64.75$_{\pm3.82}$ | 65.96$_{\pm2.52}$ |
| iTranformer | 74.14$_{\pm3.58}$ | 75.76$_{\pm2.40}$ | 75.60$_{\pm2.94}$ | 75.67$_{\pm3.38}$ |
| OneFitsAll | 62.52$_{\pm3.27}$ | 60.94$_{\pm4.43}$ | 71.30$_{\pm2.71}$ | 65.71$_{\pm3.51}$ |
| Time-LLM | 71.62$_{\pm1.65}$ | 72.56$_{\pm3.46}$ | 71.11$_{\pm4.23}$ | 71.82$_{\pm3.21}$ |
| MOMENT | 73.03$_{\pm2.63}$ | 74.55$_{\pm3.97}$ | 76.36$_{\pm2.85}$ | 75.44$_{\pm4.68}$ |
| Extra Tree Classifier | 73.95$_{\pm2.44}$ | 74.09$_{\pm2.96}$ | 75.35$_{\pm2.64}$ | 74.71$_{\pm2.03}$ |
| **Ours PhysioPFM** | **81.32$_{\pm2.65}$** | **80.77$_{\pm3.40}$** | **78.24$_{\pm2.71}$** | **79.48$_{\pm3.01}$** |

and 72.3% of valence, arousal and dominance recognition tasks, respectively. Our method benefits from the quick adaptation ability to different sub-tasks with pre-trained knowledge. In some conditions, baselines relying on large language model parameters like OneFitsAll and Time-LLM possess less knowledge of specialized downstream tasks. So, with an LLM weight as a starting point, they are inferior to other transformer-based methods such as iTtransformer.

We further investigate the performance of arrhythmia diagnosis and FoG detection on Table 3,4. For arrhythmia diagnosis, PhysioPFM leads with an average score of 89.94%, which is a notable increase over the second-best performing method by 3.39%. For FoG detection, PhysioPFM achieves SOTA performance compared to all the baselines. It shows that PhysioPFM could adapt to different downstream tasks without heavy local training.

### 5.3. Ablation Study

To further show the effectiveness of each proposed component, we conduct a comprehensive ablation study on PhysioPFM. The results shown in Table 5 demonstrate that all components contribute to the overall performance. For

the setting of ablation, we first test the validation of the anchored-classifier loss. Then we use public data with the same task as Shapelet (if we have the same domain task, else an average with 5 random ones), instead of the local data's shapelets. We also evaluate the raw TSFM without generating customized LoRA weights. The results show that ETF classifier has a great impact on model performance. A suitable temperature for contrastive learning also contributes with 1.87% to 4.63% advantages. We can also find that if the training is allowed, pre-trained TSFM could show great power in most cases after fine-tuning. Considering the variant without LoRA, we could acknowledge that the generated LoRA weights could provide a comparable improvement with direct training. The improvement from the generated LoRA enhances the pre-trained TSFM and distinguishes ours from the generic TSFM. Although shapelet has minor impacts compared with other modules, it still illustrates the importance of personalization.

**Impact of Rank.** We evaluate the impact of LoRA rank on two downstream tasks in Figure 4(a). The results show that increasing $r$ from 1 to 4 brings significant performance improvements while increasing $r$ from 8 to 16 provides no further enhancement. It shows that a larger $r$ is not neces-

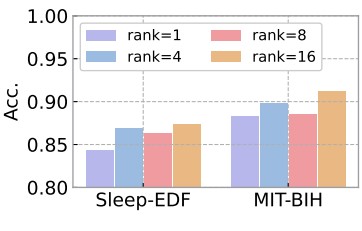
(a) Impact of LoRA rank

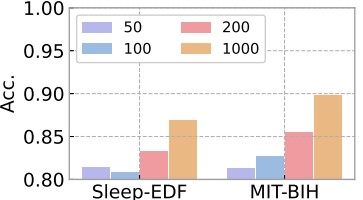
(b) Impact of DiT training sample size

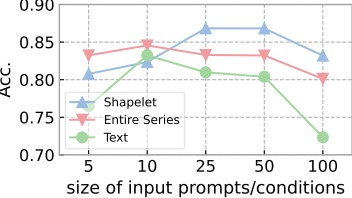
(c) Different prompts and sizes.

*Figure 4.* **PhyisoPFM capability analysis.** Experiments of (a) and (b) are conducted on sleep state detection and arrhythmia diagnosis. (c) is conducted on sleep state detection.

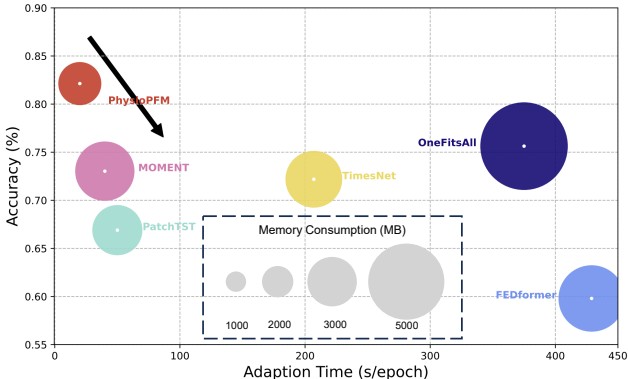

*Figure 5.* **Model performance on efficiency comparison.** The adaptation time represents the sum of training time (if needed) and inference time. We test on the FoG task.

*Table 5.* **Ablation study of PhysioPFM's modules in terms of accuracy.**

| Methods/Dataset | Sleep-EDF | MIT-BIH | FoG |
|---|---|---|---|
| TSFM with local training | 87.15 $_{\pm1.55}$ | 88.59$_{\pm1.34}$ | 83.71$_{\pm2.25}$ |
| Ours w/o $\mathcal{L}_{ac}$ | 82.47$_{\pm2.01}$ | 84.30$_{\pm1.52}$ | 78.56$_{\pm3.78}$ |
| Ours $\mathcal{L}_{ac}$, $\tau = 0.1$ | 84.64$_{\pm1.92}$ | 86.10$_{\pm2.16}$ | 79.45$_{\pm3.96}$ |
| Ours $\mathcal{L}_{ac}$, $\tau = 10$ | 81.76$_{\pm2.31}$ | 85.52$_{\pm2.84}$ | 78.37$_{\pm1.60}$ |
| Ours w/o local shapelets | 85.94$_{\pm2.05}$ | 87.43$_{\pm1.65}$ | 79.96$_{\pm3.21}$ |
| Ours w/o local LoRA | 82.08$_{\pm2.37}$ | 84.17$_{\pm2.42}$ | 76.45$_{\pm3.04}$ |
| **Our PhysioPFM** | 86.39$_{\pm2.54}$ | 89.94$_{\pm1.35}$ | 81.32$_{\pm2.65}$ |

sarily better, and the size and difficulty of the dataset matter. Smaller $r$ can also achieve acceptable performance. While a larger $r$ represents the increase in the size of parameters that LoRA needs to train and generate. A rank that balances computational overhead and performance is better.

**Impact of training samples.** We consider different sizes of samples for the generator training. Here, the training samples denote the sub-task number, i.e., the number of the map of the task and LoRA weights. The result in Table 4(b) shows that as the number of training samples increases, the performance of the model also improves. More training on bringing the link between conditions and weights contributes to the ability of the diffusion transformation. However, the number of map items relies on collecting public physiological signals.

**Training text and series as prompts.** In the original design of PhysioPFM, the shapelets are used for the prompts. We

investigate the different prompts as input in Table 4(c). In a lower prompt size, shapelets have comparable performance with the whole time series sequence. While the whole time series representation may be more complex, and inputting more prompts could have a negative impact on learning the mapping relationship between time series and LoRA weights. For texts, without robust pre-trained language encoders for time series like CLIP, the difficulty of learning is greater for textual prompts.

### 5.4. Adaptation Efficiency

As a critical issue for medical practice, we conduct a computing consumption study for baselines in the FoG task. In Figure 5, we observe three-dimensional performance: accuracy, adaptation time, and memory costs. Here, we define the adaptation time. For local training-based methods, the training time and inference time are counted for adaptation time. For other methods, it only counts the inference cost. We can find that FEDformer has the most time and computation costs. Although OneFitsAll achieves notable accuracy, training local data on the GPT-2 backbone is also time-consuming. Some lightweight models could train and predict quickly, but lose accuracy. Our PhysioPFM conducts generator training on the server, which requires about 20GB of GPU memory, which is feasible for servers with sufficient computing power. In local adaptation, DiT inference only occupies 3 GB of GPU memory, reaching a remarkable balance between accuracy, speed, and Consumption.

## 6. Conclusion

In this work, we introduce PhysioPFM, a physiological signal foundation model for train-once-for-all personalization. PhysioPFM obtains powerful neural network diffusion performance by pre-training on public medical time series. By simply inputting local private sensitive medical time series, customized LoRA weights can be generated without training, supporting the efficient personalization of time series foundation models. With the advantages of low overhead, privacy protection and efficient performance, PhysioPFM proposes a new framework for clinical diagnosis of medical physiological signals.

## Acknowledgments

This work was supported by the National Key R&D Program of China (2022ZD0160703), the National Natural Science Foundation of China (62202422 and 62372408).

## Impact Statement

This paper presents work whose goal is to advance the field of Machine Learning. There are many potential societal consequences of our work, none of which we feel must be specifically highlighted here

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

# A. Appendix

## A.1. Offline Shapelet Discovery

Considering that most existing shapelets mining methods require online training, that is, training the classification model while improving the mining quality, and our method does not need to directly use shapelets as a training feature but only as an input condition for the generator, we adopted an existing Offline Shapelet Discovery (Le et al., 2022; 2024)

In the shapelet extraction stage, OSD identifies shapelet candidates using Perceptually Important Points (PIPs). It starts by including the first and last points of the time series in the PIPs set. Then, it repeatedly adds the point with the highest reconstruction distance to the set. Each time a new PIP is added, new shapelet candidates are created using three consecutive PIPs. The number of PIPs determines the size of candidates.

In the shapelet selection stage, OSD selects shapelets by ensuring an equal number is chosen for each class. For each shapelet candidate, we calculate its Pairwise Shapelet Distance (PSD), denoted as:

$$PSD(X, S_i) = \min_{j=1}^{T-l+1} \left( \text{CID}(T[j : j + l - 1], S_i) \right) \tag{12}$$

where $X$ is the given series of length $T$, a subsequence $S_i = s_1, ..., s_l$ of length $l$, with $l \leq T$, CID is the complexity-invariant distance. This distance is used to compute the "information gain," which measures how well a shapelet separates classes. The shapelets with the highest information gain are selected and added to the final shapelet pool. To compress the input dimension, we set a fixed number $n$ for all classes $\{\mathbf{S}_i^c\}_{i=1}^n \ \forall c \in [C]$ as shapelet prototypes.

## A.2. Downstream Tasks

**Sleep-state detection.** Sleep is crucial for human health. It accounts for one-third of a human lifetime. Effective diagnosis and treatment of patients with sleep-related disorders is currently a pressing and intensively researched topic in the healthcare community (Zhang & Wu, 2017; Dong et al., 2017).

**Emotion detection.** Emotion detection is the process of identifying human emotions and has been applied in many fields such as healthcare, safe driving, and the metaverse (Ayata et al., 2020). With the rapid development of portable wearable physiological signals have been conveniently monitored and analyzed online throughout the day (Pan et al., 2023).

**Arrhythmia diagnosis.** Cardiovascular disease has become the leading cause of death from non-communicable diseases worldwide, with more people dying from cardiovascular disease each year than from any other cause of death. Arrhythmia is a common cardiovascular syndrome. Abnormal origin of cardiac electrical stimulation, abnormal conduction sequence, and frequency changes can all cause arrhythmia, which is a major challenge currently faced by the clinical treatment of cardiovascular disease. Different types of abnormal heart rhythms will show different states in the electrocardiogram waveform and frequency. Studying electrocardiogram signals can diagnose a variety of arrhythmia symptoms.

**Freezing of Gaits Detection.** Freezing of gait (FoG) is a common symptom of Parkinson's disease (PD) and usually involves an inability to cope with concurrent cognitive, limbic, and motor inputs, resulting in movement disruption (Tăuțan et al., 2020). Current clinical FoG assessments are self-report diaries by patients and manual video analysis by experts (Pham et al., 2017).

## A.3. Detailed Algorithm

We describe the detailed algorithm of the diffusion transformer training process, as followed Algorithm 1:

## A.4. Details of Baselines

We introduce the baselines for comparison:

**SimMTM** (Dong et al., 2023), a pre-trained framework on time series for recovering masked time points via weighted aggregation of multiple neighbours outside the manifold.

**TimesNet** (Wu et al., 2023), analyzes time series changes from a multi-periodicity perspective, expands one-dimensional time series data to two-dimensional space, and uses advanced visual backbone networks for feature extraction.

**iTtransformer** (Liu et al., 2024a), this method embeds the entire time series of each variable into tokens independently.

---

**Algorithm 1** LoRA generator training

---

1: **Input:** Number of training runs $N$, training sample map $\mathcal{T} = \{(\Delta W_k, t_k)\}_{k=1}^{K}$, diffusion process length $J$, diffusion cumulative variance schedule $\bar{\alpha}$.
2: **Initialize:** Learnable parameters $\phi$ for $G$
3: **for** $i = 1, 2, ..., N$ **do**
4:      # Sample a mini-batch of data from data map
5:      $(\Delta W_k, t_k) \sim \mathcal{T}$
6:      # Noise LoRA parameters
7:      $j \sim U(\{1, ..., J\})$
8:      $\Delta \tilde{W}_k^j \sim \mathcal{N}(\sqrt{\bar{\alpha}_j} \Delta W_k, (1 - \bar{\alpha}_j)I)$
9:      # Compute the predictions
10:     $\Delta \hat{W}_k \leftarrow G_\phi(\mathbf{S}(t_k), \Delta \tilde{W}_k^j, j)$
11:     # Compute the loss
12:     loss $\leftarrow ||\Delta \hat{W}_k - \Delta W_k||_2^2$
13:     # Update DiT's parameters
14:     $\phi_{i+1} \leftarrow$ update(loss; $\phi$)
15: **end for**

---

**PatchTST** (Nie et al., 2023), processes each dimension of the multivariate time series separately, that is, inputs each dimension into the Transformer Backbone separately, and then splices the obtained prediction results along the dimension direction.

**FEDformer** (Zhou et al., 2022), reduces the distribution shifts between input and output through seasonal-trend decomposition; and proposes a model structure that applies an attention mechanism in the frequency domain to increase robustness to noise.

**Informer** (Zhou et al., 2021), proposes ProbSpare Self-Attention effectively replaces the traditional Self-Attention.

**OneFitsAll** (Zhou et al., 2023), trains time series data on the pre-trained GPT-2 weights.

**Time-LLM** (Jin et al., 2024), freezes the LLM backbone, and then uses two learnable modules (Patch Reprogramming and Output Projection) to reprogram the input and output respectively.

**MOMENT** (Goswami et al., 2024). collects a large and diverse collection of public time series, Time-series Pile. Then it uses a lightweight reconstruction head to reconstruct the embedding of the input time series for pre-training.

**SleepFM** (Thapa et al., 2024), curates a large polysomnography dataset of multi-modal sleep recordings and trains the model by contrastive learning.

**SleepDG** (Wang et al., 2024a), designs an epoch-level feature alignment to align the feature distribution of each single sleep epoch between different domains and designs a sequence-level feature alignment to minimize the difference in sequence features between different domains.

**LSTM-MLP** (Wang et al., 2023), proposes an emotion recognition method based on feature fusion of single-lead EEG and ECG signals using various time domain, frequency domain and nonlinear features.

**OMHGL** (Pan et al., 2023), includes multimodal hypergraph fusion and online hypergraph learning. Multimodal hypergraph fusion can fuse multimodal physiological signals and effectively obtain emotion-related information. Online hypergraph learning aims to learn new information from online data by updating hypergraph projection.

**DeepArr** (Midani et al., 2023), combines feed-forward and recurrent deep neural networks using a sequential fusion approach to exploit relevant feature representations of arrhythmias in electrocardiogram (ECG) signals.

**Extra Tree Classifier** (Goel et al., 2023), combines the integration technology of multiple prediction methods to improve the performance of the EEG signal gait freezing detection model

### A.5. Evaluation Metrics

In this section, we describe the evaluation metrics we adopted in the experiments for classification tasks. We first introduce the concept of a confusion matrix. TP (True Positive): a correctly predicted positive example. That is, the true value of the

data is a positive example, and the predicted value is also a positive example; TN (True Negative): a correctly predicted negative example. That is, the true value of the data is a negative example, and the predicted value is also a negative example; FP (False Positive): a wrongly predicted positive example. That is, the true value of the data is a negative example, but it is wrongly predicted as a positive example; FN (False Negative): a wrongly predicted negative example. That is, the true value of the data is a positive example, but it is wrongly predicted as a negative example.

**Accuracy.** Accuracy refers to the ratio of correctly classified samples to the total number of samples.

$$Accuracy = \frac{TP + TN}{TP + FP + TN + FN}. \tag{13}$$

**Precision.** Precision refers to the proportion of samples that are positive among the samples predicted to be positive.

$$Precision = \frac{TP}{TP + FP}. \tag{14}$$

**Recall.** Recall refers to the ratio of the actual number of positive samples in the predicted positive samples to the total number of positive samples in the samples.

$$Recall = \frac{TP}{TP + FN}. \tag{15}$$

**F1.** The F1 score is a weighted average of precision and recall.

$$F1 = 2\frac{Precision \times Recall}{Precision + Recall}. \tag{16}$$

**Macro F1.** The Macro F1 score refers to the average of each class's F1 score.

$$Macro - F1 = \frac{1}{N}\sum_{i=1}^{N} F1_i \tag{17}$$

**Kappa coefficient**. $\kappa$ is a method used to evaluate the consistency of classification models.

$$
\begin{aligned}
P_0 &= \frac{TP + TN}{TP + FP + TN + FN} = Accuracy \\
P_e &= \frac{(TP + FP) \times (TP + FN) + (FN + TN) \times (FP + TN)}{(TP + TN + FP + FN)^2} \\
\kappa &= \frac{P_0 - P_e}{1 - P_e}
\end{aligned}
\tag{18}
$$

**AUC.** AUC (Area Under the Curve) is an indicator for evaluating the performance of binary classification models, usually used as the area under the ROC curve (Receiver Operating Characteristic curve). AUC represents the ability of the classifier to rank positive examples before negative examples.

$$\text{AUC} = \int_0^1 \text{Precision}(\text{Recall}^{-1}(u))du \tag{19}$$

### A.6. Public Datasets

In Table 6, we provide statistical information on collected public physiological signals, mainly from (Zhang et al., 2024; Qiu et al., 2023) and PyhsioNet (Goldberger et al.). We exclude the datasets for experiment evaluations from pre-training.

## B. Theoretical analysis

The proof for Proposition 3.1 is given as:

*Table 6.* **Statistics of the physiological benchmark datasets.**

| Task | Name | Modalities | # of subjects | Sampling rate |
|---|---|---|---|---|
| Arrhythmia Diagnosis | MIT-BIH arrhythmia dataset (Moody & Mark, 2001) | ECG | 1 | 360Hz |
|  | PTB-XL (Wagner et al., 2020) | ECG | 71 | 500Hz |
|  | European ST-T database (Taddei et al., 1992) | ECG | 2 | 250Hz |
|  | AF classification challenge 2017 (Clifford et al., 2017) | ECG | 4 | 300Hz |
|  | PTB diagnostic ECG (Wagner et al., 2020) | ECG | 9 | N/A |
|  | AHA (Moody & Mark, 1982) | ECG | 8 | 250Hz |
|  | CPSC2018 (Liu et al., 2018) | ECG | 8 | 500Hz |
| Denoise | MIT-BIH noise stress test (Moody et al., 1984) | ECG | 1 | 360Hz |
| Emotion Detection | CLAS (Markova, 2020) | ECG,BVP,EMG,GSR | 60 | 256HZ |
|  | SWELL (Koldijk et al., 2014) | ECG,SC | 25 | 2048HZ |
|  | ASCERTAIN (Subramanian et al., 2016) | EEG, ECG, EDA | 58 | N/A |
|  | BIO-VID-EMO DB (Zhang et al., 2016) | ECG, EMG, SC | 86 | N/A |
|  | DREAMER (Katsigiannis & Ramzan, 2017) | EEG, ECG | 23 | 256Hz |
| Seizure Detection | Hospital (TUH) (Obeid & Picone, 2016) | EEG | 315 | 200Hz |
| Sleep-state Detection | Sleep-EDF (Kemp et al., 2000) | EEG, EOG, EMG | 22 | 100Hz |
|  | SHHS (Zhang et al., 2018) | EEG, EOG, EMG | 5804 | 12550Hz |
|  | ISRUC (Khalighi et al., 2016) | EEG, ECG, EOG, EMG | 118 | 200Hz |
|  | HMC (Alvarez-Estevez & Rijsman, 2021) | EEG, ECG, EOG, EMG | 151 | 256Hz |
| Motor Imagery | BCI Competition IV (Blankertz et al., 2006) | EEG, EOG | 9 | 250Hz |
|  | EEG Motor Movement/Imagery Dataset (Schalk et al., 2004) | EEG, EOG | 109 | 160Hz |
| Freezing of Gaits | Li *et. al* (Li, 2021) | EEG, EMG, ECG, SC, ACC | 12 | 1000Hz |
| Cyclic Alternating Pattern | CAP (Terzano et al., 2001) | EEG, ECG, EOG, EMG | 108 | 512Hz |
| Sleep Apnea | MIT-BIH PSG (Ichimaru & Moody, 1999) | EEG, ECG, EOG, EMG | 18 | 250Hz |
|  | UCDDB (Goldberger et al., 2000) | EEG, ECG, EOG, EMG | 25 | 128Hz |
| Stress and Affect Detection | WESAD (Schmidt et al., 2018) | ECG, EMG | 15 | 700Hz |

*Proof.* Suppose $C$ class prototype vectors $p_1, p_2, \ldots, p_C \in \mathbb{R}^d$ satisfy: Unitization: $|p_i| = 1$, symmetric inner product constraint: $p_i^T p_j = -\frac{1}{C-1}, \forall i \neq j$. Define the Gram matrix $G \in \mathbb{R}^{C \times C}$, where:

$$G_{ij} = p_i^T p_j = \begin{cases} 1 & \text{if } i = j, \\ -\frac{1}{C-1} & \text{if } i \neq j. \end{cases}$$

The matrix has the following properties: the diagonal elements are 1 and the off-diagonal elements are $-\frac{1}{C-1}$. The matrix rank is $d$, and it is a symmetric semi-positive matrix (because the vector is in d-dimensional space). The above Gram matrix corresponds to the Simplex Equiangular Tight Frame. Its core features are: all off-diagonal elements are equal, that is, the angles between vectors are consistent, the vectors are evenly distributed in the feature space, and the minimum interval between classes is maximized. This structure is the only configuration that satisfies symmetry and minimizes the similarity between classes. The angle between any two vectors $\theta$ satisfies:

$$\cos \theta = p_i^T p_j = -\frac{1}{C-1}.$$

In classification problems, the decision boundary is determined by the geometric relationship of the prototype vectors. For a linear classifier, the decision boundary for two categories $i$ and $j$ is:

$$\left\{ x \in \mathbb{R}^d \mid (p_i - p_j)^T x + \frac{\|p_j\|^2 - \|p_i\|^2}{2} = 0 \right\}.$$

Since $\|p_i\| = \|p_j\| = 1$, the boundary is simplified to:

$$(p_i - p_j)^T x = 0.$$

The margin between the two boundaries is:

$$\text{Margin} = \frac{2}{\|p_i - p_j\|}.$$

Calculate $\|p_i - p_j\|^2 = 2(1 - p_i^T p_j) = 2\left(1 + \frac{1}{C-1}\right)$, so:

$$\text{Margin} = \frac{2}{\sqrt{2\left(1 + \frac{1}{C-1}\right)}} = \sqrt{\frac{2(C-1)}{C}}.$$

When all class prototypes meet the symmetry condition, the intervals between all classes are consistent and reach the maximum possible value, so the decision boundary is optimal.

