# OpenReview forum: "Efficient Personalized Adaptation for Physiological Signal Foundation Model"
_ICML.cc/2025/Conference — ICML 2025 poster_

### Official Review · Reviewer_57t7 · 2025-03-07

**Overall Recommendation:** 3

**Summary:**

This work proposes a new method to adapt physiological signal foundation models using DiT to generate LoRA weight matrices.

**Claims And Evidence:**

The authors claim that the proposed method transfer physiological foundation model to different tasks with lower computing costs. While it is true that the proposed method claims to not need any training during adaptation stage, the proposed method requires significant compute to obtain the LoRA dataset and train the DiT. On the other hand, the baseline methods do not need any training between the pre-training and adaptation stage. In essence, the proposed method requires front-loading the adaptation compute to the stage between pre-training and adaptation. I think the author should tune down the claim for lower adaptation costs and be clear about the extra training compute required to obtain the LoRA dataset and train the DiT. Furthermore, it could be preferable to compare the total compute used after pre-training stage, it can be argued that the LoRA dataset preparation and DiT training is part of adaptation.

**Essential References Not Discussed:**

What about GAN-based or hypernetworks for generating the LoRA weights? Why was DiT selected for generating the weights of LoRA when GAN and hypernetworks can achieve a similar purpose. Prior works have shared the similar idea of generating weights using a neural network for adaptation, perhaps these are relevant baselines to consider, since the core contribution of this work is generating LoRA weights using DiT.

**Experimental Designs Or Analyses:**

How about a baseline that fine-tunes the pre-trained TSFM (6-layer GPT2-based backbone). While I understand that this violate data privacy, all the baseline methods use different backbone networks. This will help to understand how much of the base performance comes from the pre-trained TSFM and how much improvement the DiT generated LoRA provides.
Aslo consider a prototypical network as baseline? During adaptation, one can use only the pre-trained TSFM (6-layer GPT2-based backbone), without finetuning LoRA, to calculate the prototypes, and make predictions via metric-based classifier. This is very compute efficient adaptation. This will also give an idea of how good the pre-trained TSFM is, as well as how much improvement the DiT generated LoRA provides.
ECG-FM was mentioned in related works, but not benchmarked for ECG arrhythmia task.

**Methods And Evaluation Criteria:**

The benchmarks are reasonable.

**Other Comments Or Suggestions:**

Table 1 Weak
Section 3.3 multimodel
Section 4.2 depp learning

**Other Strengths And Weaknesses:**

None.

**Questions For Authors:**

Section 4.1 Experimental setup, it was claimed that 60:20:20 train:val:test split was performed. How is the 60% training data used for conditioning the DiT, i.e. were all 60% of data used for shapelet discovery? How is the validation set used, if at all?
The 60:20:20 split was randomly sampled, data from one patient can be present in both the training set and validation/test split. What is the “personalized” adaptation? It appears to be capable of adapting to unseen dataset.

**Relation To Broader Scientific Literature:**

The idea of using one neural network to generate weights for another neural network (hypernetworks) is not itself novel, but it has not been done with DiT to the best of my knowledge.

**Theoretical Claims:**

No theorem or proof was provided.

---

> ### Author Rebuttal · Authors · 2025-04-01
>
> Thank you for your extensive feedback. We especially appreciate your effort to meet the conference's rigorous review criteria. We kindly address your questions as follows.
>
>
> **W1:** Scope of adaptation time.
>
> **For W1:** Thanks for your thoughtful comments. We separated the two phases of pre-training and adaptation, considering a practical cloud-edge scenario. If utilizing a traditional pre-trained TSFM, the TSFM is first pre-trained with massive data in a venue with sufficient resources (cloud). Then, the TSFM is used to be trained on local physiological signals, and its training time is the adaptation time we defined. We did not count the original pre-training of TSFM into its adaptation time. Similarly, for our method, pre-training can be achieved using public physiological signals and a venue with sufficient resources. Our adaptation time includes generator inference and TSFM inference. Therefore, we respectfully disagree that LoRA dataset preparation and DiT training are part of adaptation time.
>
>
> **W2:** Discussion of important ablation study and baselines.
>
> **For W2:** We sincerely thank you for your detailed suggestions.  We would like to address your concerns by showing additional ablation results for our method. We adopt the TSFM with local LoRA fine-tuning, TSFM with a prototype-based classifier, ECG-FM on arrhythmia diagnosis, along with two given row results in Table 5 as references. From the results, we can find that if the training is allowed, pre-trained TSFM could show great power in most cases after fine-tuning. Considering the variant without LoRA, we could acknowledge that the generated LoRA weights could provide a comparable improvement with direct training.  A prototypical network is also a valuable approach as it could surpass general baselines but still be lower than the adapted weight-based strategy (Ours, fine-tuning TSFM). It may require more computing cost in similarity(metric) calculation, and could be easily affected by imbalanced data. More alignment or neural-collapse tricks may help to improve this approach. Metric learning is a promising future work to explore. We will also add the complete results of ECG-FM to Table 3 in the revised manuscripts.
>
> | Method   | Sleep-EDF | MIT-BIH | FoG
> |---| --- | --- | --- |
> | TSFM with local training     | 87.15    |  88.59   |  83.71   |
> | TSFM with Proto-classifier |  82.24     |   81.06   |  74.56  |
> | ECG-FM | -  | 84.90  | - |
> | Ours w/o local LoRA | 82.08 |84.17| 76.45 |
> | Ours | 86.39| 89.94 | 81.32  |
>
> **W3:** Clarifications on generator selection.
>
> **For W3:** In parameter generation tasks, on the one hand, these small models may not be able to fully demonstrate their generalization ability when applied to more complex tasks and parameter spaces. On the other hand, previous methods do not support conditional high-performance parameter generation, and the novel DiT-based conditional neural network diffusion has better generation results. The architecture of DiT has great expressive power in diffusion tasks, especially in conditional cross-modal applications such as text-to-image and text-to-video. Unlike hypernetwork, which takes the model parameters as input and generates parameters, we directly map the data feature space to the parameter space. In addition, the hypernetwork needs to perform backpropagation based on the loss of the backbone network, which will be costly in large model scenarios. We conducted a validation experiment, taking the condition and noised model parameters as input, and measuring the Euclidean distance between the output model parameters and the original input parameters. The results show that the generation effect of the method based on conditional GAN or MLP is far behind DiT. As the SOTA cross-modal generator, we adopted DiT. It is also worthwhile to find the trade-off between cost and quality in future work.
> | Similarity   | DiT | MLP | CGAN |
> |---| --- | --- | --- |
> | Task 1     | 4.10    |  153.52   |  79.63  |
> | Task 2 |  2.32     |  139.46   |  75.49  |
>
> **W4:** Typos.
>
> **For W4:** Thank you for this astute observation. We apologize for any confusion caused and have carefully revised it.
>
> **W5:** Clarifying the data partition and definition of personalization.
>
> **For W5:** The experiments are conducted on corresponding signals in a subject-independent setting. We assign subjects to train/val/test partitions. One subject’s data won’t appear across train/val/test sets. The inference of the generator is based on testing sets, where the train and val sets are not used for our method but for baselines, for fairness. Compared with the generalized TSFM, our personalization refers to obtaining a model parameter that adapts to local data for new local tasks and data features. Because the generalized model may not work well on specific tasks, and fine-tuning a large foundation model will also be costly, realizing model personalization in a lightweight way is meaningful.

---

> > ### Comment · Reviewer_57t7 · 2025-04-04
> >
> > Thanks for the clarifications. I have updated my recommendation.
> >
> > I understand the adaptation time explanation. While the adaptation phase is more compute efficient, I still think it would be valuable to provide a comparison of the computational requirements for traditional pre-training versus the DiT training proposed in this work. This will give readers an idea of how much computation is required for cloud training phase.
> >
> > I recommend the remaining clarifications to be included in the final manuscript, as they provide important information and context.

---

> > > ### Author Response · Authors · 2025-04-04
> > >
> > > Thank you for your reply. We're delighted to have addressed your concerns and appreciate your helpful suggestions. We agree with the importance of the cost of two-phase cloud pre-training and the remaining clarifications. We promise to revise all of these points in the manuscript.

---

### Official Review · Reviewer_qTAF · 2025-03-08

**Overall Recommendation:** 3

**Summary:**

* This paper studies medical time series classification based on physiological signals.
* ML models for prediction from medical time series is challenging since we often have:
    * Unbalanced amount of data for each signal
    * Varying sampling frequency/duration
* Time series foundation models (TSFMs) can be effective, however they can be expensive to adapt to individual medical centres, and we may not want to upload data from a given centre to a server for finetuning.
* This work proposes an approach where we:
    * Learn a dataset of LoRA weights for a TSFM
    * Learn a diffusion transformer to output LoRA weights for a pretrained TSFM, using the LoRA weights found above as a training set, with the DiT input being a condensed representation  of the input dataset (shapelet transformed)
* At inference time, for a new dataset, can run inference with shapelet + diffusion model to get LoRA weights, then adapt the TSFM without any further training cost.
* The method is evaluated on four classes of physiological signal datasets: Sleep state detection, emotion detection, arrhythmia diagnosis, and freezing of gait detection.
* The proposed model performs well, improving on baselines.

**Claims And Evidence:**

Overall reasonable when compared to baselines.

**Essential References Not Discussed:**

Not familiar enough with literature to comment here.

**Experimental Designs Or Analyses:**

Positives:
* Diverse range of datasets and tasks studied
* Good selection of published baselines
* Ablations generally thorough
* Overall, encouraging results compared to the baselines.

Areas for improvement/questions:
* I would like more detail on the evaluation datasets. How are these split into train/val/test? Is it patient level? Without this, it’s hard to interpret the results well.
* What kind of hyperparameter search happened with your method, vs the baselines?
* Why diffusion for the LoRA learning? Could the standard MLP used in eg hypernetworks work, especially given small size of parameters? Would that be easier to train?
* What happens if you take your pretrained TSFM and do local LoRA without your diffusion model at all? I understand this is computationally expensive, but it’s interesting to understand the effect of generative model for LoRA vs just learning the weights directly.

**Methods And Evaluation Criteria:**

Evaluation datasets are sensible; methods could use further investigation, see below.

**Other Comments Or Suggestions:**

Overall, I think this is a good contribution, but the lack of related work in the main body makes it hard to contextualise the work. I also have some questions above that would be great to get clarity on.  If these are answered, I would be inclined to increase my score.

**Other Strengths And Weaknesses:**

* The writing needs work overall — there are a number of typographical errors, referencing a Table instead of a Figure, for example
* The overall contribution of the method was not that clear. Rewording to more closely represent the Figure 3 would be quite valuable.

**Questions For Authors:**

Please see above.

**Relation To Broader Scientific Literature:**

The related work in the main paper is inadequate. A more detailed related work in main body would add a lot of value, in addition to what is in the appendix.

**Theoretical Claims:**

Not major focus of paper.

---

> ### Author Rebuttal · Authors · 2025-04-01
>
> We sincerely thank you for your valuable comments, and we are grateful for the time and effort you have invested in reviewing our work. Below, we provide a point-by-point response to address each of your concerns:
>
> **W1:** Clarifying the data partition.
>
> **For W1:** Thanks for your valuable comments. It is at the patient level. Physiological signal data are collected from individual subjects. The experiments are conducted on corresponding signals in a subject-independent setting. We divide the subjects into training, validation, and test sets in a 3:1:1 ratio. One subject’s data won’t appear across train/val/test sets. It may arouse concerns about how it can be considered a personalized adaptation (for baselines) without training on the data of a specific local patient. In practice, local test data must be unlabeled, and the model training is through existing labeled data, and then the model is applied to the patients who need to be diagnosed. So, this setting is reasonable and in line with the existing related works' paradigm.
>
> **W2:** Hyperparameter search.
>
> **For W2:** We test the best learning rate, diffusion steps, chunk size and rank of the adapter for our methods. For baselines, we generally follow the original hyperparameters and keep necessary settings fair, such as data partition.
>
> **W3:** Discussion on the setting and selection of the generator.
>
> **For W3:** The architecture of DiT has great expressive power in diffusion tasks, especially in conditional cross-modal applications such as text-to-image and text-to-video, where the original DiT is applied. Small models like MLP have difficulty in cross-modal generation and conditional generation. In difficult tasks such as parameter generation, the latent space generated by MLP may be poorly representative. Therefore, how to achieve a trade-off between the lightness and performance of the generator is a future work worth exploring. In this work, we mainly applied the SOTA cross-modal generator DiT to achieve high-quality parameter generation. At the same time, the transformer complies with the scaling law, a feature that will help the model to generalize only by amplifying parameters. We conduct an analysis of the generation quality on weights, considering the Euclidean distance between input weights(noised) and generated weights. As demonstrated in the following results, at the end of the training of DiT, the results of distance reached 2.32 to 4.10 for different tasks, proving superior generating performance than conditional GAN and MLP. In addition, MLP and CGAN require huge numbers of rounds to converge, more than 1000 epochs, while DiT does not.
> | Similarity   | DiT | MLP | CGAN |
> |---| --- | --- | --- |
> | Task 1     | 4.10    |  153.52   |  79.63  |
> | Task 2 |  2.32     |  139.46   |  75.49  |
>
>
> **W4:** Detailed ablation study on model fine-tuning.
>
> **For W4:** We sincerely thank you for your insightful feedback.  We evaluate the TSFM with local LoRA fine-tuning, and the results are in the following table.  We also add the existing results from Table 5, which includes ours without LoRA generation, and our method. It can be seen that training TSFM on local data could achieve superior performance in most cases. While our generated LoRA weights could also provide a comparable improvement to direct training. Without training or generated weights, the pre-trained TSFM is hard to fit the specific domain knowledge well.
>
> | Method   | Sleep-EDF | MIT-BIH | FoG
> |---| --- | --- | --- |
> | TSFM with local training     | 87.15    |  88.59   |  83.71   |
> | Ours w/o local LoRA | 82.08 |84.17| 76.45 |
> | Ours | 86.39| 89.94 | 81.32  |
>
>
> **W5:** Typo and presentations.
>
> **For W5:** We appreciate the reviewer's keen eye and thorough review. We have carefully revised the typos and will provide a more rigorous expression of Figure 3 in the revised version of our work.
>
> **W6:** Suggestions on related work.
>
> **For W6:** Thanks for your constructive suggestions. Due to the length limit, we have to adopt the current organization. It is truly significant to place the related work section in the main body for better understanding by the readers. We promise to reorganize it. We will also be adding more extensive related work, such as privacy protection-related work.

---

### Official Review · Reviewer_FR4e · 2025-03-12

**Overall Recommendation:** 3

**Summary:**

The paper provides a personalized approach to transfer the time series foundation model to clinical physiological signal tasks. The main constraints are the lower computing costs and privacy.

**Claims And Evidence:**

Not always. A main issue is that it is not clear how the authors are addressing the privacy protection in the paper. The paper would have benefitted from a clear positioning within the related literature. See comments bellow for more details.

**Essential References Not Discussed:**

There are some missing related work. Since this paper seeks to address privacy issues, it would have been relevant to include some related work on this topic.

**Experimental Designs Or Analyses:**

The experimental designs and the conducted analysis seem to be acceptable, and the ablation study demonstrates the relevance of the chosen techniques.

**Methods And Evaluation Criteria:**

Yes

**Other Comments Or Suggestions:**

-

**Other Strengths And Weaknesses:**

A major issue in this paper is the lack of clear description of the contributions within the privacy constraint. The paper states that its major contribution is privacy preserving. However, there is no much information on the privacy, and how it is clearly preserved. Moreover, the paper does not cite any related work from the privacy preserving literature.

It is not clear from the paper that Definition 2.1 is a contribution  or not.
Moreover, the paper and appendix do not present the proof of Proposition 3.1. It is not clear how it was established.

The two main ingredients in this work are (i) to train the foundation model with massive public physiological time series by using a low-rank adapter, and (ii) to use a neural collapse in order to combat the imbalance in the data distribution.
Moreover, a diffusion transformer is used as a robust generator to synthesize the low-rank adapter weights.

**Questions For Authors:**

-

**Relation To Broader Scientific Literature:**

The paper roughly integrates several concepts within a single foundation model to pre-train/fine-tune it. This paper provides some interesting results, and the ablation study confirms the relevance of the chosen techniques, such as low-rank adapter and neural collapse.

**Theoretical Claims:**

There is no proof given for Proposition 3.1.

---

> ### Author Rebuttal · Authors · 2025-04-01
>
> We sincerely appreciate for your comprehensive and valuable review, particularly given the meticulous standards during the review of this venue. We appreciate the opportunity to address your concerns.
>
> **W1:** Clarification on the privacy issue.
>
> **For W1:** We apologize for missing a detailed discussion on privacy preservation. Privacy constraint is one of our contributions. A generic pre-trained time series foundation model is hard to fit the diverse local tasks and data. Uploading patients’ data to the cloud for robust pretraining may raise privacy concerns. Our proposed method achieves this goal by data isolation, that is, sensitive patients’ data are preserved locally.  Due to the removal of data exposure, privacy issues could be solved. The most related work includes some time series foundation models and federated learning. Brant-X [1] adopts the EEG foundation model Brant-2 as a basis, which is pre-trained on 4TB of private brain signal data. This approach requires private data and large-scale pre-training on it. If the medical entity has sufficient computing resources, it could be reasonable, otherwise, it may need to transfer private data to a robust cloud. On the other hand, federated learning preserves privacy by exchanging models, not data, which is similar to our paradigm.  [2] considers using FL to train a foundation model for medical time series. Multiple rounds of exchange on the large models may lead to remarkable communication costs, and models may be attacked by model poisoning attacks from history gradient updates. While our work focuses on the personalized TSFM with a privacy guarantee.  Due to the length limitation of the rebuttal, we promise to add a more comprehensive privacy-preserving literature on related work and our method.
>
> [1] Zhang, Daoze, et al. "Brant-X: A Unified Physiological Signal Alignment Framework." KDD 2024.
>
> [2] Ali, Mahad, et al. "Fine-Tuning Foundation Models with Federated Learning for Privacy Preserving Medical Time Series Forecasting." IEEE EMBC 2025.
>
> **W2:** Definition 2.1 and Proposition 3.1.
>
> **For W2:** Thanks for your valuable comments. Definition 2.1 follows the standard definition of neural collapse. It is not a contribution, but an illustration.
>
> The proof for Proposition 3.1 is given as:
>
> $\textit{Proof.}$ Suppose $C$ class prototype vectors $p_1, p_2, \dots, p_C \in \mathbb{R}^d$ satisfy: Unitization: $|p_i| = 1$, symmetric inner product constraint: $p_i^T p_j = -\frac{1}{C-1}, \forall i \neq j$. Define the Gram matrix $G \in \mathbb{R}^{C \times C}$, where:
> $$
> G_{ij} = p_i^T p_j =
> \begin{cases}
> 1 & \text{if } i = j, \newline
> -\frac{1}{C-1} & \text{if } i \neq j.
> \end{cases}
> $$
>
> The matrix has the following properties: the diagonal elements are 1 and the off-diagonal elements are $-\frac{1}{C-1}$. The matrix rank is $d$, and it is a symmetric semi-positive matrix (because the vector is in d-dimensional space). The above Gram matrix corresponds to the Simplex Equiangular Tight Frame. Its core features are: all off-diagonal elements are equal, that is, the angles between vectors are consistent, the vectors are evenly distributed in the feature space, and the minimum interval between classes is maximized. This structure is the only configuration that satisfies symmetry and minimizes the similarity between classes.
> The angle between any two vectors $\theta$ satisfies:
> $$
> \cos\theta = p_i^T p_j = -\frac{1}{C-1}.
> $$
> In classification problems, the decision boundary is determined by the geometric relationship of the prototype vectors. For a linear classifier, the decision boundary for two categories $i$ and $j$ is:
> $$
> x \in \mathbb{R}^d \mid (p_i - p_j)^T x + \frac{\|p_j\|^2 - \|p_i\|^2}{2} = 0.
> $$
>
> Since $\|p_i\| = \|p_j\| = 1$, the boundary is simplified to:
> $$
> (p_i - p_j)^T x = 0.
> $$
>
> The margin $\gamma$ between the two boundaries is:
> $$
> \gamma= \frac{2}{\|p_i - p_j\|}.
> $$
>
> Calculate $\|p_i - p_j\|^2 = 2(1 - p_i^T p_j) = 2\left(1 + \frac{1}{C-1}\right)$, so:
> $$
> \gamma = \frac{2}{\sqrt{2\left(1 + \frac{1}{C-1}\right)}} = \sqrt{\frac{2(C-1)}{C}}.
> $$
> When all class prototypes meet the symmetry condition, the intervals between all classes are consistent and reach the maximum possible value, so the decision boundary is optimal.
>
> We will add this proof to the appendix in the revised manuscripts.

---

> > ### Comment · Reviewer_FR4e · 2025-04-08
> >
> > Dear authors,
> > I would like to thank you for your feedback.
> > However, the rebuttal is strengthening my major concerns that this work is not well positioned in the literature, mainly federated learning literature. There are many papers that address federated learning for time series, such as (and many papers more recent):
> > - Zhuang, W., Chen, C., & Lyu, L. (2023). When foundation model meets federated learning: Motivations, challenges, and future directions. arXiv preprint arXiv:2306.15546.
> > For healthcare, see for instance the following paper and references within:
> > - He, Y., Huang, F., Jiang, X., Nie, Y., Wang, M., Wang, J., & Chen, H. (2024). Foundation model for advancing healthcare: challenges, opportunities and future directions. IEEE Reviews in Biomedical Engineering.
> >
> > For all these reasons, and taking into account the feedback provided, as well as the other reviews, I will maintain my scores.

---

> > > ### Author Response · Authors · 2025-04-08
> > >
> > > Dear Reviewer FR4e,
> > >
> > > Thanks for your response. We apologize for making you feel that not well-positioning this work in the literature. We would like to kindly clarify the relation to federated learning. Our work considers a cloud-edge scenario, where a cloud (cloud computing platform or AI company) with sufficient computing resources. It is able to train and provide pre-trained foundation models with public general time series data and physiological signals. Considering the clinical site as an edge, we aim to design a lightweight TSFM adaptation approach to personalize the received TSFM without exposing local patient data. For federated learning, it considers a collaborative training scenario with a cloud server and multiple clients. In FL, each local client is able to train the model. While fine-tuning a foundation model in the client is still costly, even with LoRA or other techniques. Many works are also exploring the effectiveness of exchanging LoRA, where direct exchange has performance loss compared to the original entire model. Multiple communication requires extra cost and is vulnerable to model poisoning attacks. Therefore, federated learning with foundation models tries to address the mentioned challenges, which are different from ours. In short, we consider transforming generic TSFM to personalized in a local training-free view, while FL focuses on collaborative training effectively and privacy-preserving. We regard them are orthogonal but somehow related in a privacy view.
> > >
> > > We sincerely thank you for your constructive suggestions. We promise to try our best to improve the related work literature and position of the privacy issue to be clearer in revising the manuscript. Our main contribution still lies in the techniques of efficient weight generation to boost the large model's lightweight personalization.
> > >
> > > Best regards,
> > >
> > > Authors

---

### Official Review · Reviewer_Q918 · 2025-03-14

**Overall Recommendation:** 4

**Summary:**

This paper proposes a novel approach to achieve efficient adaptation for physiological signal foundation modals to private datasets. The main idea is to use Low-rank Adaptation (LoRA). However, unlike existing methods that train LoRA weights for adaptation, it utilizes a diffusion model to generate the LoRA weights. To this end, the paper first prepared 30 datasets and obtained LoRA weightings for them. They are then used to train a diffusion model with a Transformer as the backbone. The shapelet prototypes are extracted from the datasets and used as conditions in the diffusion generation. The trained diffusion model can then be used to generate a LoRA weight for private data. Empirical evaluations are performed on four typical physiological classification tasks.

**Claims And Evidence:**

The paper claims that the proposed PhysioPFM is more robust and efficient than existing approaches utilizing a time series foundation model for physiological signal tasks. The claim is well supported by the empirical evaluations. Comparison with recent SOTA methods indicates the effectiveness of the proposed approach. The efficiency comparison also shows that the proposed PhysioPFM is more efficient in terms of memory consumption and adaption time.

**Essential References Not Discussed:**

No.

**Experimental Designs Or Analyses:**

The experimental settings are sound. Sufficient details of the experimental design are included, and the baselines compared are appropriate. The evaluation metrics used are suitable. An ablation study is also performed.

**Methods And Evaluation Criteria:**

The proposed method is technically sound. The use of a diffusion model to generate LoRA weight is quite interesting and novel. The benchmark datasets used are appropriate, and the evaluation metrics are also suitable.

**Other Comments Or Suggestions:**

Please see my comments above.

**Other Strengths And Weaknesses:**

Strengths:
- The proposed method of generating LoRA weights is novel and interesting.
- The paper is organized, well presented, and easy to follow.
- The experiments are well executed, and the results are pretty promising.

Weaknesses:
- The diffusion model takes only the shaplets as a condition to generate LoRA weights. This seems to be a bit restrictive for multi-task scenarios since the same LoRA weights will be generated for different tasks given the same physiological signal input.
- The diffusion model is trained using only 30 public physiological signal datasets; the impact of the number of datasets used in training the diffusion model is unclear.

**Questions For Authors:**

Usually, the inference of diffusion models is quite time-consuming. Does the adaptation time reported in Fig.5 of PhysioPFM correspond to the time needed for the diffusion model to generate the LoRA weight?

**Relation To Broader Scientific Literature:**

Fine-tuning foundation models using LoRA has been intensively explored in the literature. However, using a generative model to generate LoRA weights is a novel approach. The framework overall is similar to the hypernetwork approach [1]. The authors should also discuss the connections with the hypernetwork methods.

[1] Chauhan, V.K., Zhou, J., Lu, P., Molaei, S. and Clifton, D.A., 2024. A brief review of hypernetworks in deep learning. Artificial Intelligence Review, 57(9), p.250.

**Theoretical Claims:**

The Proposition 3.1 follows directly from Definition 2.1. No other theoretical claims are made.

---

> ### Author Rebuttal · Authors · 2025-04-01
>
> Thanks for recognizing the value of our work. We are grateful for your thorough feedback, especially considering the massive requirements of the review. We hope the following comments could address your questions:
>
> **W1:** Relation to hypernetwork.
>
> **For W1:** In a macro sense, our method and the hypernetwork both provide adaptive parameters for another backbone network. Specifically speaking, according to Figure 1(b) of the reference, the hypernetwork also needs to be trained together with the main model, calculate the performance loss of the main model, and perform gradient backward propagation on the hypernetwork. Unlike the hypernetwork, we do not update the hypernetwork based on the performance of the backbone model, which is costly in our context. Our paper directly maps the data feature space to the parameter space, aiming to establish a mapping between LoRA parameters and data features. Our method pursues better generation quality and aims to learn an effective and representative latent space. This is logically different from the traditional hypernetwork and is designed to meet the challenges of our scenario to achieve local training-free adaptation. In general, our method has similar macroscopic goals to hypernetworks, but their specific implementations are quite different. We will include a discussion of this aspect in the main text and provide more detailed related work.
>
> **W2:** Discussion of applying to multi-task scenarios.
>
> **For W2:** Thank you for raising this important concern. We focus on general physiological signals classification in this work. When the target task changes, e.g. transferring to a time series forecasting task, our approach may need to adjust the pre-training on the generator to build a new ability for new tasks. Another possible way is to expand the type of input condition, which could be a significant future work to extend our methods to multi-task scenarios.
>
> **W3:** The impact of training sample size for the generator.
>
> **For W3:** We kindly note that we have included the impact of training sample size on the generator in Section 4.3, Impact of training samples. To fully enhance the ability of the generator, we adopt multiple ways to expand the training sets in the data preparation phase, including partial dataset, partial class, and random selection of subjects. Given the different input of data, the generalized ability of the generator could be more diversified.
>
> **W4:** Definition of the adaptation time.
>
> **For W4:** We apologize for any confusion caused. For our proposed method, Figure 5 includes the time to generate LoRA weights and TSFM inference. For baselines, the adaptation time refers to the local training time. Because our method does not need to fine-tune the foundation model, while the baselines need to train and update the model. Backpropagation on a large model is time-consuming. We only have a single-step inference to generate LoRA, which takes little time compared to multiple-round training.

---

### Decision · Program_Chairs · 2025-05-01

**Decision:**

Accept (poster)

**Comment:**

This paper proposes a novel approach to achieve efficient adaptation for physiological signal foundation models to private datasets. The main idea is to using Low-rank Adaptation (LoRA). However, different from existing methods that train LoRA weights for adaptation, this paper utilizes a diffusion model to generate the LoRA weights, which is novel and interesting. All the four reviewers are positive to the work, although they also raised a lot of comments that can help to improve the work. The authors should improve the paper when preparing for the final version of the paper.